# Sleep Quality and Nutrient Intake in Japanese Female University Student-Athletes: A Cross-Sectional Study

**DOI:** 10.3390/healthcare10040663

**Published:** 2022-04-01

**Authors:** Fumi Hoshino, Hiromi Inaba, Mutsuaki Edama, Saya Natsui, Sae Maruyama, Go Omori

**Affiliations:** 1Department of Health and Nutrition, Niigata University of Health and Welfare, 1398 Shimami-cho, Kita-ku, Niigata 950-3198, Japan; inaba@nuhw.ac.jp (H.I.); whm19001@nuhw.ac.jp (S.N.); 2Athlete Support Research Center, Niigata University of Health and Welfare, 1398 Shimami-cho, Kita-ku, Niigata 950-3198, Japan; edama@nuhw.ac.jp (M.E.); omori@nuhw.ac.jp (G.O.); 3Institute for Human Movement and Medical Sciences, Niigata University of Health and Welfare, 1398 Shimami-cho, Kita-ku, Niigata 950-3198, Japan; hpm20010@nuhw.ac.jp; 4Department of Health and Sports, Niigata University of Health and Welfare, 1398 Shimami-cho, Kita-ku, Niigata 950-3198, Japan

**Keywords:** female athletes, questionnaire study, Pittsburgh Sleep Quality Index, nutrient intake

## Abstract

Objectives: This study aimed to characterize the sleep quality and nutrient intake of Japanese female college athletes to provide specific conditioning support. Methods: A cross-sectional survey conducted between December 2019 and January 2020 during the regular training season in Niigata City, located at 139°02′ E longitude and 37°55′ N latitude. Overall, 120 female university student-athletes from eight clubs were selected. All sports were at the national level of competition. The Pittsburgh Sleep Quality Index (PSQI) assessed sleep quality. Nutrient intake was assessed using Excel nutrition software through the Food Intake Frequency Questionnaire. Results: The mean PSQI score was 4.5 ± 2.5, with 29% of participants having a PSQI score ≥ 5.5. The sleep duration was 6.8 ± 1.1 h, with 45% of participants sleeping <7 h. The energy intake was 1800 ± 419 kcal, with no correlation between PSQI score and nutrient intake. Conclusions: PSQI scores were higher compared with other studies, but many participants had shorter sleep duration and lower nutrient intake, these data suggest that there is the possibility of improving the habits of Japanese athletes by increasing the subjects’ amount of sleep time and food intake.

## 1. Introduction

Sleep is essential for healthy physiological functioning. Therefore, sleep disturbance causes weight gain [1,2], hastening the onset of diseases and endangering mental health [3], which further results in secondary problems such as lack of concentration and poor academic performance. In athletes, sleep is considered to play an essential role in physical and psychological recovery [4], and inadequate sleep has been associated with muscle weakness [5], increased heart rate, ventilation, lactate production [6], and subjective exercise intensity [5] at specific exercise intensities. Furthermore, numerous previous studies have demonstrated the importance of sleep in health and performance [7,8]. Notably, a close association between sleep and nutrients has been reported, which suggests that a high-carbohydrate diet improves sleep onset latency compared with a low-carbohydrate diet. This study also proposes that a high-protein diet reduces wake episodes [9]. Additionally, it has been reported that middle-aged Japanese women with poor sleep quality have a lower intake of green and yellow vegetables and a higher intake of confectionary [10].

Therefore, athletes should maximize their performance by paying careful attention to their sleep and dietary habits. Some reports have investigated the relationship between sleep quality and nutrient intake status in athletes [11,12,13]. Additionally, college student-athletes should manage their own life and nutrition to improve their athletic performance and continue their athletic life in the future. Nevertheless, a lack of data has been observed on sleep and nutrition education for college student-athletes.

Furthermore, several previous investigations regarding the sleep patterns of elite athletes exist [4,14,15,16]. However, a lack of data evaluating the characteristics of college athletes and considering specific nutrition status measures persists. Additionally, only a few reports have been identified that simultaneously investigate sleep status and nutrient intake in college student-athletes. College athletes should develop self-management skills and maintain/improve their health during tournaments to continue competing and prevent future health impacts.

We hypothesized that college student-athletes with poor sleep quality would have a low intake of nutrients, fruits, and vegetables and a higher intake of confectionary, on the basis of which this survey was conducted. This study investigated the sleep and nutrient intake status of Japanese female college student-athletes with the aim of providing specific future support to improve sleep and nutrition habits.

## 2. Materials and Methods

### 2.1. Ethical Considerations

The study protocol was approved by the Ethics Committee of the Niigata University of Health and Welfare (No. 18032-180723). This study complies with the Helsinki Declaration. The study contents were fully explained to the subjects. Written informed consent was obtained from all subjects.

### 2.2. Participants

A total of 120 female college athletes from eight clubs (swimming, track and field sprinting, track and field long-distance, track and field throwing and jumping, soccer, basketball, volleyball, and creative dance) were included in the cross-sectional study. A self-administered questionnaire was administered. All club athletes competed at the national level. The study period was between December 2019 and January 2020 and was conducted during the regular training season. The survey was conducted in the city of Niigata, which is located at 139°02′ E longitude and 37°55′ N latitude.

### 2.3. Anthropometry

Height (cm), body weight (kg), and body fat (%) were measured using a body composition monitor (DC150, TANITA, Tokyo, Japan). Body mass index (BMI) was calculated as body weight (kg)/height (m^2^).

### 2.4. Questionnaire

This study used a questionnaire to measure sleep quality and nutrient intake. The sleep quality of the participants was assessed using the Pittsburgh Sleep Quality Index (PSQI) [17]. The questionnaire inquired about the participants’ sleep status during the past month. It comprised 18 questions, categorized into seven factors: sleep quality, sleep duration, sleep onset time, sleep efficiency, sleep difficulties, use of sleep-inducing drugs, and interference with daily life due to daytime sleepiness. Subsequently, each element was given a score of 0–3 points, following which the total and PSQI scores (0–21) were calculated. In this study, sleep was considered disturbed when the PSQI score was above the cutoff of 5.5 points [18].

Furthermore, the frequency of food intake questionnaire (FFQg version 4.0; Kenpaku Co., Ltd., Tokyo, Japan) was also used to survey nutrient consumption with the Excel nutrition software (v.7.0, Kenpaku Co., Ltd., Tokyo, Japan) [19]. Nutrient intake and food group intake was calculated per kg of body weight.

### 2.5. Statistical Analysis

Of the 120 participants, those with missing responses were excluded. Ultimately, 112 participants were included in the analysis. Data are presented as mean ± standard deviation. Based on the PSQI score, the two groups were divided, one with no risk of sleep disturbance and the other with risk, and the nutrient intakes were compared between the two groups. Correlations between PSQI scores and nutrient intakes were determined. Statistical processing was performed using SPSS Statics Version 28 for Windows, and a significant probability of less than 5% was considered statistically significant.

## 3. Results

### 3.1. Participants

The primary attributes of the participants are shown in Table 1. The participant’s age was 19.8 ± 1.0 years, height was 162.7 ± 6.1 cm, weight was 57.5 ± 9.1 kg, BMI was 21.7 ± 2.7 kg/m^2^, and body fat percentage was 22.0% ± 4.1%.

### 3.2. Sleep Quality and Sleep Disturbances

The PSQI scores of the participants (Table 2), the distribution of PSQI scores (Figure 1), the distribution of scores by category (Figure 2), and the distribution of sleep duration (Figure 3) are shown in the figures. The overall PSQI score for the participants was 4.5 ± 2.5. Thirty-three participants (29%) were determined to be at risk for sleep disorders. The distribution of the total PSQI scores showed a tendency for many of them to be in the 3-point range. Of the seven categories, high scores were found in four: C1: sleep quality, C2: sleep onset time, C3: sleep duration, and C7: difficulty waking up during the day. The duration of sleep was 6.8 ± 1.1 h.

### 3.3. Nutritional Intake

Nutrient and food group intake is shown in Table 3. Energy intake was 1800 ± 419 kcal. The balance of energy-producing nutrients was 14% protein, 34% fat, and 52% carbohydrate. The PSQI scores were used to compare the intake of nutrients and food groups in the no-risk and at-risk sleep disorder groups. No significant differences in nutrient intake between the two groups were found. The no risk for sleep disorder group had a significantly higher intake of beans than the with risk group.

## 4. Discussion

This study assessed Japanese female college athletes’ sleep and nutrient intake habits.

Our study showed that the mean PSQI score of the participants was 4.5 ± 2.5. However, elite Japanese female athletes reported a PSQI score of 4.7 ± 2.2 [16]. In contrast, while the American female college student-athlete reported a mean PSQI score of 5.3 ± 2.5 [20], the Japanese female college student-athletes at other universities reported a score of 5.8 ± 2.5 [21]. Moreover, the general Japanese female college student population recorded a score of 7.53 ± 2.24 [22]. Results also showed 29.3% of the participants as having a PSQI score of 5.5 or higher. On the other hand, the percentages of elite Japanese female athletes, American college student-athletes, other Japanese female college student-athletes, and general Japanese college students with 5.5 or higher PSQI scores were 32.1% [16], 42.4% [20], 48% [21], and 79.2% [22], respectively. Additionally, the sleep scores of the participants were lower than those of other studies, even for those with a PSQI of 5.5 or higher. Therefore, our results suggest that many participants had better sleep conditions than in other reports.

The average sleep duration of the participants in this study was 6.8 h. In contrast, elite Japanese female athletes reported 7.3 h of sleep [16], American college student-athletes reported 6.98 h [20], Japanese female college student-athletes at other universities reported 6.1 h [21], and the general Japanese female college students reported 6.3 h [22]. Furthermore, although the participants’ average sleep duration was longer than that of other Japanese college athletes and general college students, it was similar to that of American college athletes and shorter than that of elite Japanese female athletes. The university included in this investigation was located in a rural area of Japan and did not require a long commute. This reason is proposed to explain why the participants included in this investigation were able to get more sleep than other university students in Japan. It is recommended that the adult population sleep for at least 7 hours per day to maintain good health [23]. However, in this study, 45% of the participants slept less than 7 h. It has also been reported that sleep duration affects the performance of athletes [24,25], making it necessary to increase the participants’ sleep duration. Also worth noting was that PSQI score and sleep duration varied among the sports clubs included in this study. It has been reported that early morning practice limits the level of sleep an athlete can obtain [14,15]. Therefore, the variation in the PSQI scores and sleep duration of each club is proposed to be because of practice time, especially early morning practice. Nevertheless, we did not investigate the training period or whether morning practice was used in this study. Therefore, it is necessary to investigate the effects of these influences in future investigations.

Since the scores of the participants were high in C1: sleep quality, C2: time to fall asleep, C3: sleep duration, and C7: difficulty waking up during the day on the PSQI, a specific way to improve the participants’ sleep situation would be by improving these factors.

The participants were divided into two groups according to their PSQI scores (the good and bad sleep groups). Furthermore, although nutrient intake was compared between the two groups, no significant difference was observed. The correlation between nutrient intake and PSQI score was also determined and no correlation was observed. Low energy intake reduces the quality of sleep, which is proposed to have influenced the results of this investigation [7]. The subjects’ mean energy intake was recorded to be 1800 kcal, which is lower than the 1950 kcal [26] required by the average woman of the same age. Additionally, their protein intake per kg body weight was 1.1 g, which was lower than the 1.2–2.0 g per kg body weight recommended for athletes [27,28]. Therefore, since a high-protein diet has been reported to reduce wake episodes [9], it is possible that increasing the subjects’ energy and protein intake can improve their sleep status. Based on intake by the food group, it has been reported that those who consume milk more frequently have better subjective sleep quality [29]. Others have also reported that those with higher PSQI scores have a lower intake of green and yellow vegetables [10,30]. Several recent reports have examined the relationship between sleep quality and nutritional status in athletes [11,12,13]. Sleep and nutrient intake studies in male elite athletes using wrist activity monitors have reported the following. Evening protein intake was associated with shortened sleep onset latency, and evening sugar intake was associated with shorter total sleep time and wake time after sleep onset [11]. Similar studies in female athletes have also reported the following. High carbohydrate intake increases wake time after sleep onset and high iron intake increases sleep duration [12]. Another study of endurance athletes aged 18–60 years also reported higher sleep disturbance scores with higher daily caffeine intake [13]. Hence, we also compared the quantity of each food group and nutrient intake between the two groups (good and bad sleep groups), but no significant difference was observed. In this study, only a questionnaire was used to measure sleep status and a wrist activity monitors were not used. It may be necessary to use a wrist activity monitor or to survey sleep and nutrient intake for multiple days. In summary, although the PSQI scores of the female college student-athletes in this study were better than those in other studies, the average of sleep duration was 6.8 h and their nutrient intake was low. Nutrient and food group intake was compared between the two groups; no significant difference was observed. It has also been pointed out that in Japan, little coverage of sleep in school education exists [31]. Therefore, it is necessary to implement education and environmental improvement related to sleep and education on both sleep and diet in the future.

## 5. Limitations

The sleep state measurement in this study was a questionnaire-based survey and not an objective determination via a polygraph or actigraph. Thus, the presence or absence of a sleep disorder cannot be determined solely by the results of the PSQI score. Additionally, although depression was reported to affect sleep status [32], we did not investigate mental status in this study. Therefore, the results were not adjusted for such factors. Furthermore, although this study examined the current status of college female athletes, we excluded non-athletes. Consequently, it was impossible to compare the subjects’ situation with other university students studying at the same university. Moreover, the results of this study cannot be applied to male athletes. Additionally, there have been reports regarding the influence of gender differences in the subjective perception of sleep quality [33,34,35]. Therefore, it is necessary to conduct separate surveys for men and women to understand their characteristics and consequently make specific interventions based on the findings. Finally, although the food intake frequency survey estimates intake based on the frequency of dietary intake in the past, it is necessary to consider that over- or under-reporting can occur because the data are self-described by the subjects.

## 6. Conclusions

This study simultaneously investigated Japanese female college athletes’ sleep quality and nutrient intake. Nutrient and Food group intake was compared between the two groups, no significant difference was observed. PSQI scores were good compared to other reports, but the average amount of sleep was 6.8 h, shorter than the recommended 7 h for maintaining good health. In addition, low energy and protein intake was observed. Therefore, these data suggest that there is the possibility of improving the habits of Japanese athletes by increasing the subjects’ amount of sleep time and food intake.

## Figures and Tables

**Figure 1 healthcare-10-00663-f001:**
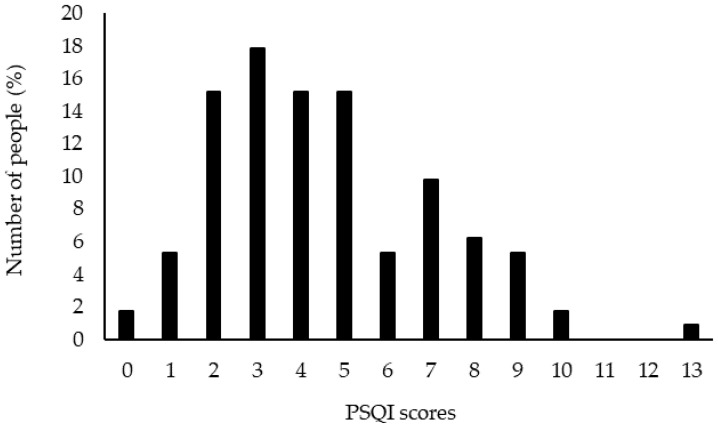
Distribution of PSQI scores, *n* = 112.

**Figure 2 healthcare-10-00663-f002:**
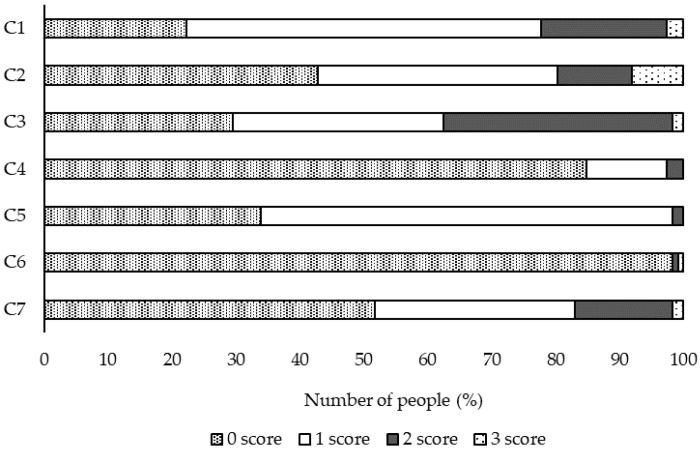
Distribution of scores for each component of the Pittsburgh Sleep Questionnaire, *n* = 112.

**Figure 3 healthcare-10-00663-f003:**
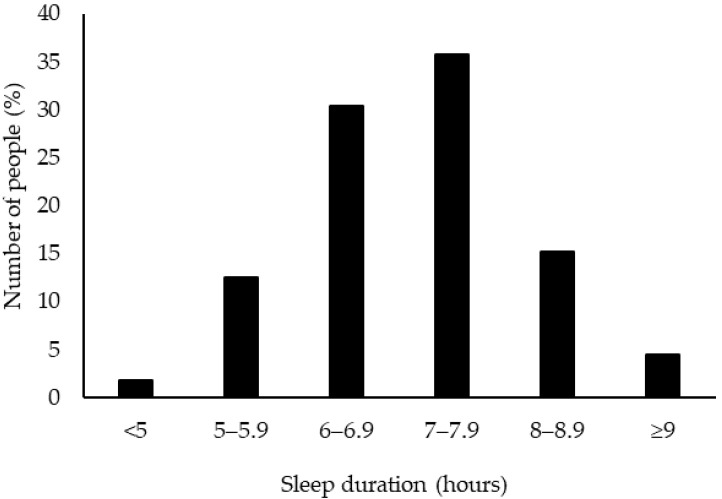
Distribution of Sleep Duration, *n* = 112.

**Table 1 healthcare-10-00663-t001:** Physical characteristics of the subjects, *n* = 112.

Subject characteristics		Total (*n* = 112)	Pittsburgh Sleep Quality Index Score		*p*-Value
			Good Score (*n* = 79)	Poor Score (*n* = 33)	
PSQI global score	score	4.5 ± 2.5	3.2 ± 1.3	7.8 ± 1.5	-
Age	years	19.8 ± 1.0	19.7 ± 1.0	19.9 ± 1.0	0.347
Height	cm	162.7 ± 6.1	163.5 ± 6.3	160.7 ± 5.4	0.027
Weight	kg	57.5 ± 9.1	58.5 ± 8.1	55.2 ± 10.7	0.077
BMI	kg/m^2^	21.7 ± 2.7	21.8 ± 2.2	21.3 ± 3.6	0.366
Body Fat	%	22.0 ± 4.1	22.0 ± 4.2	22.2 ± 4.0	0.800
Sleep duration	hours	6.8 ± 1.1	7.0 ± 1.1	6.2 ± 0.9	<0.001

PSQI: Pittsburgh Sleep Quality Index, BMI: body mass index. Mean ± SD.

**Table 2 healthcare-10-00663-t002:** The PSQI scores of the subjects, *n* = 112.

Sleep Status		Total *n* = 112	Swimming *n* = 15	Athletic Sprint *n* = 18	Athletic Long-Distance *n* = 6	Athletic Throwing/Jumping *n* = 9	Soccer *n* = 17	Dance *n* = 13	Basketball *n* = 20	Volleyball *n* = 14
PSQI	score	4.5 ± 2.5	3.9 ± 2.3	4.8 ± 2.5	5.5 ± 2.0	5.0 ± 3.0	3.6 ± 2.5	5.7 ± 3.0	4.5 ± 2.2	4.4 ± 2.6
Bedtime	hh:mm	0:14	0:04	0:05	23:25	0:46	23:30	0:41	0:37	0:35
Wake-up time	hh:mm	7:26	7:44	7:31	6:05	7:25	7:23	7:37	7:12	7:48
Sleep duration	hours	6.8 ± 1.1	7.4 ± 0.6	6.8 ± 1.0	6.6 ± 0.8	6.3 ± 1.4	7.4 ± 1.3	6.3 ± 0.7	6.3 ± 1.0	6.8 ± 1.2
PSQI >5.5	%	29	20	44	33	33	31	46	25	14

Mean ± SD.

**Table 3 healthcare-10-00663-t003:** Nutrient intake and Food group intake of the subjects, *n* = 112.

Energy, Nutrients and Foods		Total (*n* = 112)	Pittsburgh Sleep Quality Index Score		
			Good Score (*n* = 79)	Poor Score (*n* = 33)	*p*-Value
Energy	kcal	1800 ± 419	1835 ± 443	1717 ± 345	0.174
Energy	kcal/kg B.W.	32 ± 11	32 ± 8	33 ± 16	0.567
Protein	g/kg B.W.	1.2 ± 0.4	1.1 ± 0.4	1.2 ± 0.6	0.237
Fat	g/kg B.W.	1.2 ± 0.5	1.2 ± 0.4	1.3 ± 0.6	0.406
Carbohydrate	g/kg B.W.	4.0 ± 1.4	3.9 ± 0.9	4.1 ± 2.1	0.720
Ca	mg/kg B.W.	8.7 ± 4.1	8.3 ± 3.5	9.6 ± 5.3	0.148
Fe	mg/kg B.W.	0.12 ± 0.06	0.1 ± 0.0	0.1 ± 0.1	0.138
Retinol equivalent	μgRAE/kg B.W.	8.5 ± 4.5	8.3 ± 3.6	9.1 ± 6.2	0.379
Vitamin D	μg/kg B.W.	0.08 ± 0.06	0.08 ± 0.06	0.09 ± 0.06	0.326
α tocopherol	mg/kg B.W.	0.10 ± 0.05	0.10 ± 0.04	0.11 ± 0.06	0.345
Vitamin B_1_	mg/kg B.W.	0.02 ± 0.01	0.02 ± 0.01	0.02 ± 0.01	0.247
Vitamin B_2_	mg/kg B.W.	0.02 ± 0.01	0.02 ± 0.01	0.02 ± 0.01	0.340
Vitamin B_6_	mg/kg B.W.	0.02 ± 0.01	0.02 ± 0.01	0.02 ± 0.01	0.326
Vitamin B_12_	mg/kg B.W.	0.09 ± 0.06	0.09 ± 0.06	0.10 ± 0.06	0.411
Vitamin C	mg/kg B.W.	1.3 ± 0.9	1.2 ± 0.7	1.4 ± 1.4	0.396
Dietary fiber	g/kg B.W.	0.2 ± 0.1	0.2 ± 0.1	0.2 ± 0.2	0.292
Grains	g/kg B.W.	6.05 ± 2.33	6.00 ± 1.67	6.18 ± 3.47	0.779
Potatoes	g/kg B.W.	0.47 ± 0.52	0.42 ± 0.42	0.61 ± 0.70	0.144
Green and yellow vegetables	g/kg B.W.	1.01 ± 0.86	0.96 ± 0.69	1.11 ± 1.19	0.418
Other vegetables	g/kg B.W.	1.94 ± 1.38	1.86 ± 1.00	2.14 ± 2.02	0.451
Seaweed	g/kg B.W.	0.07 ± 0.07	0.06 ± 0.07	0.07 ± 0.07	0.632
Beans	g/kg B.W.	0.99 ± 1.00	0.86 ± 0.89	1.30 ± 1.19	0.034
Fish	g/kg B.W.	0.76 ± 0.74	0.73 ± 0.75	0.84 ± 0.74	0.505
Meats	g/kg B.W.	2.10 ± 1.08	2.08 ± 1.02	2.15 ± 1.23	0.751
Eggs	g/kg B.W.	0.62 ± 0.39	0.62 ± 0.40	0.64 ± 0.37	0.784
Dairy products	g/kg B.W.	2.13 ± 1.78	2.12 ± 1.67	2.16 ± 2.04	0.904
Fruits	g/kg B.W.	1.12 ± 1.58	1.09 ± 1.38	1.17 ± 1.99	0.814
Sweets and snacks	g/kg B.W.	1.19 ± 0.79	1.17 ± 0.78	1.22 ± 0.83	0.771
Sugar-sweetened beverages	g/kg B.W.	1.47 ± 1.85	1.52 ± 1.77	1.34 ± 2.04	0.652
Sugar	g/kg B.W.	0.08 ± 0.06	0.08 ± 0.07	0.07 ± 0.06	0.515
Nuts and seeds	g/kg B.W.	0.03 ± 0.08	0.03 ± 0.10	0.02 ± 0.03	0.479
Fats and oils	g/kg B.W.	0.25 ± 0.13	0.25 ± 0.13	0.25 ± 0.13	0.886
Seasoning and spices	g/kg B.W.	0.59 ± 0.37	0.62 ± 0.39	0.53 ± 0.31	0.228

Mean ± SD.

## Data Availability

The data that support the findings of this study are available from the corresponding author.

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
