# Peer review of "Sleep Quality and Nutrient Intake in Japanese Female University Student-Athletes: A Cross-Sectional Study"

_healthcare, 2022, doi:10.3390/healthcare10040663_

Round 1

Reviewer 1 Report

Dear authors,

The subject is of interest, and there are certainly not many published studies on the matter, and even less with data on Japanese athletes and with such an interesting sample size. However, there are some considerations that I think could be improved:

1. According to the STROBE guidelines, it is important to explicitly identify the study design in the title, abstract, and methods section. Although it is understood to be a cross-sectional, I have not been able to find it anywhere.

2. Unless the journal rules indicate otherwise, the abstract of the article should be structured (background, objectives, methods, results, conclusions).

3. Introduction: this section is well described, places the reader on the subject and shows a knowledge gap, although three studies are lacking as previous research: https://pubmed.ncbi.nlm.nih.gov/32624442/; https://www.frontiersin.org/articles/10.3389/fspor.2022.810402/full and https://pubmed.ncbi.nlm.nih.gov/34952801/

4. Materials and methods: the study design is not stated.

5. Results:  adequately described.

6. Discussion: it is missing to compare the results with three studies: https://pubmed.ncbi.nlm.nih.gov/32624442/; https://www.frontiersin.org/articles/10.3389/fspor.2022.810402/full and https://pubmed.ncbi.nlm.nih.gov/34952801/

7. Conclusions: the final sentence does not correspond to the results of this research: "Therefore, these data suggest that sleep conditions could be improved by increasing the subjects' amount of sleep time and food intake", a conclusion that cannot be reached with this research. It is proposed to modify by "Therefore, these data suggest that there is the possibility of improving the habits of Japanese athletes by increasing the subjects' amount of sleep time and food intake".

Reviewer 2 Report

General Comment: The article presents current relevance, the introduction reflects current knowledge and the methodology used was adequate and well described. The results are correctly presented and the discussion is sufficiently substantiated. However, the use of more up-to-date references should be considered, particularly in the discussion section.

Comment #1:

Line 42: The reference to middle-aged women is outside the scope of the article. Clarification must be made.

Line 61: According to the paper, the aim must refer to the fact that the study assessed Japanese, female, college student-athletes. Correction must be made.

Line 106: BMI must be presented with units, kg/m2. Correction is in order.

Line 118: The following statement "the majority of those were 6–7 hours of sleep." is not in agreement with data shown on Figure 3. Clarification is in order.

Line 136: Given the description of the use of correlations and once conclusions are drawn in this regard, we suggest that these data should be presented. Otherwise, no conclusions should be drawn about correlations.

Line 139: Table 4 is a continuation of Table 3, therefore it should be presented as such. Correction must be made.

Lines 144, 145 and 149: PSQI Score mean value lacks Standard Deviation. Correction must be made. 

Line 226: The conclusion "...sleep duration was shorter than the recommended 7 hours for..." is not supported by the fact that 55 % of the subjects presented a sleep duration > 7h (as Shown on Figure 3 and stated on line 178. Clarification is in order.

Comment #2: In the Reference section, 25 of the articles used are prior to 2017, therefore, with more than 5 years old, and of these, 4 of them are prior to the year 2000. An update of the bibliography is recommended.
